# Un'ontologia per il patrimonio culturale teatrale su Wikibase.cloud

*Donatella Gavrilovich gavrilovich@lettere.uniroma2.it*
*Giovanni Bergamin giovanni.bergamin@logos-ri.eu*
*Valeria Paraninfi valeria.paraninfi@uniroma2.it*

## 1. Introduzione

Le basi di dati e di conoscenza, dedicate alle arti dello spettacolo e concepite sui principi dell'*Open Science,* sono oggi ampiamente presenti in rete. Alcuni dei prodotti più significativi sono "AusStage" (2003), il più grande *gateway* digitale al mondo, il portale francese "Les Archives du Spectacle" (2007), "ECLAP" (European Collected Library of Artistic Performance, 2010), "Swiss Performing Arts Platform" (2012), "Frankfurt Performing Arts for theatre and dance studies" (2015), il "National Portal of Theatre Museums of Russia" (2019), "Digital Pina Bausch Archive" (2020).
"AusStage" è stato il modello per la creazione della piattaforma norvegese "IbsenStage", lanciata nel 2014 (Hanssen 2018, 24), così come "ECLAP" lo è stato per le successive realizzazioni degli archivi digitali europei, soprattutto per quanto riguarda la struttura ontologica e la scheda digitale dei metadati progettati sulla base di standard internazionali. Ciò nonostante, ECLAP non ha soddisfatto la committenza e il sito è stato disattivato dopo pochi anni. Il limite di ECLAP è dovuto proprio alla sua struttura ontologica, che raccoglie e suddivide, per categorie di appartenenza, gli elementi eterogenei che costituiscono l'evento teatrale. Essa obbliga l'utente ad avviare sempre una nuova ricerca, perché il sistema fornisce la connessione dati senza un ordine chiaro e di facile utilizzo. Inoltre, la scheda dei metadati è molto concisa e non esiste alcuna connessione con altri elementi dello stesso evento.

In Italia non esiste né un museo nazionale teatrale, né una piattaforma che colleghi tutti gli archivi teatrali digitali esistenti, pubblici e privati, per conoscere, condividere e valorizzare il patrimonio dei beni teatrali delle singole istituzioni italiane. Studiosi, archivisti e professionisti delle arti dello spettacolo hanno manifestato la necessità di creare una piattaforma comune per esigenze di ricerca e di lavoro (Gavrilovich 2020). Il Ministero della Cultura (MiC) ha accolto questa esigenza a suo modo e nel dicembre 2020 ha pubblicato online "Gli Archivi dello Spettacolo", un database che raccoglie i metadati essenziali del contenuto di archivi, posti sotto la sua tutela, ordinandoli per regioni. Si tratta di una sorta di inventario, che nasce dalla necessità del MiC di conoscere ciò che tutela come "patrimonio di particolare rilevanza storica", ma che non risponde ai diversificati fabbisogni dell'utenza.

## 2. Il progetto Hyperstage: origini e obiettivi per una nuova base di conoscenza

Catalogare lo spettacolo teatrale significa non solo raccogliere in modo ordinato, secondo gli standard vigenti, un'enorme quantità di dati eterogenei riguardanti documenti e materiali (ad es. copioni, costumi, bozzetti di scena, figurini, spartiti, fotografie, note di regia ecc.), ma anche creare un collegamento logico tra essi, avendo come scopo la ricostruzione filologica dell'unità storico-artistica e socio-culturale di una rappresentazione teatrale che non esiste più.

Per raggiungere questo risultato bisogna partire dalla concezione unitaria dello spettacolo teatrale, che deve essere considerato come un'opera d'arte totale, allo stesso modo di un'opera d'arte figurativa complessa qual è, ad esempio, un polittico. Per questo motivo nella creazione della struttura ontologica di una base di conoscenza dedicata alle arti dello spettacolo è necessario procedere in modo che gli elementi costituenti di quel tutto, che non esiste più, tornino a "dialogare" come in origine, incrociando i dati. (Gavrilovich 2017, 28-39).

Partendo da questi concetti e riflessioni, è stato progettato nel 2014 un prototipo per una base di conoscenza *Performance Knowledge base* (PKB) dedicata alla organizzazione dei dati delle produzioni teatrali (Gavrilovich 2017, 28-39). Dieci anni dopo questa sperimentazione è confluita nel progetto PRIN 2022 *HYPERSTAGE: an Open Knowledge base for the semantic reconstruction of theatrical performances through the harvesting and processing resources from the New Italian Network of theatrical digital archives*[1].

La principale innovazione è la metodologia del progetto Hyperstage, basata su competenze e conoscenze umanistiche. Essa ha permesso di affrontare e risolvere il problema, finora irrisolto da parte degli esperti IT (Diwisch, e Thull 2014) (Estermann 2017; 2020) ( Beck, e Voß 2020) (Weiberg 2020), della contestualizzazione storica, sociale, politica, artistica e culturale delle informazioni inerenti una certa rappresentazione teatrale, così come dell'organizzazione dell'enorme quantità dei suoi dati eterogenei.

Applicando le metodologie solitamente in uso nell'ambito degli studi teatrali (De Marinis 2008), è stato possibile trovare la giusta soluzione che si basa, da un lato, sull'organizzazione dei dati secondo le tre fasi fondamentali della produzione di un'opera teatrale (concezione, realizzazione e post-produzione); dall'altro, sulla raccolta strutturata per tipologie dei materiali che sono stati ordinati in due raggruppamenti distinti (Gavrilovich 2013; 2017; 2020). Il primo gruppo include tutti gli oggetti digitali caratterizzati da immagini; il secondo gruppo include quelli caratterizzati da testo. Nei casi dubbi, ad esempio una locandina con testo e immagine, il criterio da seguire è la rilevanza dell'uno rispetto all'altro.
Hyperstage introduce un'importante innovazione nel mondo Wiki: la classificazione PKb grazie alla quale è possibile gestire la grande quantità e l'eterogeneità delle risorse digitali e di cui parleremo in seguito.

## 3. Preservare l'effimero: un'ontologia per le arti teatrali

### 3.1 Necessità di una ontologia

---

[1] Le unità di ricerca del progetto Hyperstage sono: Unità di Ricerca 1 [Università di Roma "Tor Vergata", Associazione Logos-ri di Firenze e Consortium GARR di Roma], Unità di Ricerca 21 [ Università di Bologna], Unità di Ricerca 3 [Accademia di Belle Arti di Venezia]. I partner fornitori di contenuti sono: Teatro Stabile di Torino, Teatro alla Scala di Milano, Accademia Olimpica di Vicenza, Fondazione Romaeuropa e Fondazione INDA di Siracusa. Donatella Gavrilovich è il Principal Investigator.

Per creare una base di conoscenza aperta (Open knowledge base) dedicata alla rappresentazione semantica degli spettacoli teatrali, è necessario strutturare i dati utilizzando un'ontologia di dominio compatibile con il Web semantico. In informatica, un'ontologia definisce le entità, i loro attributi e le relazioni tra le entità, favorendo l'interoperabilità e la condivisione della conoscenza in un dominio specifico[2].

Quando si presenta una ontologia può essere d'aiuto ricordare la fondamentale distinzione tra il livello TBox e il livello ABox (Nardi, e Brachman 2010, 16-20). Il livello TBox (dove 'T' sta per 'terminological') definisce il vocabolario dell'ontologia: le entità di un determinato dominio (classi e proprietà) e i vincoli che regolano i collegamenti tra queste entità. Possiamo dire anche che il livello TBox si occupa dello schema o del modello dei dati che decidiamo di trattare. Il livello ABox (dove 'A' sta per 'assertional') si occupa invece dei 'dati' che popolano il modello o lo schema definito nel livello TBox. Gli elementi che caratterizzano questo livello sono gli individui (o le specifiche istanze) delle classi definite nel TBox, le caratteristiche e le relazioni specifiche tra questi individui.

Il dominio teatrale manca di un'ontologia consolidata e generalmente adottata. Recenti studi esplorano lo sviluppo di ontologie per le arti dello spettacolo (performing arts). Ad esempio Mitsopoulou, Kyprianos, e Brattis (2024) analizzano modelli come: lo Swiss Performing Arts - SPA - data Model (Estermann, e Schneeberger 2017); l'iniziativa Linked digital future della Canadian Arts Presenting Association, CAPACOA (Estermann, e Julien 2019); il Wikiproject Performing arts in ambito Wikidata[3]. Un altro studio (Skaug, e Aalberg 2024) - dedicato ai modelli concettuali collegati alla documentazione degli spettacoli teatrali (theater performances) e in particolare al concetto di opera (concept of work) - prende in considerazione anche le strutture di metadati che non si basano su un'ontologia formale - come ad esempio IbsenStage.

Per favorire l'interoperabilità, framework generalisti come Wikidata e Schema.org sono importanti, ma non specifici per il teatro. In particolare Schema.org è stato creato per essere un vocabolario minimo comprensibile dai motori di ricerca, mentre Wikidata si propone di essere un nodo centrale (un hub) tra ontologie di dominio. La principale differenza tra Schema.org e Wikidata è che la prima è una ontologia definita a livello di TBox da una comunità basata su un accordo tra i principali motori di ricerca[4], mentre Wikidata - attiva dal 2012 - è sia una base di conoscenza aperta, sia una ontologia basata sulla collaborazione (Samuel 2017): in altre parole sia il livello TBox che il livello ABox vengono sviluppati mettendo in pratica il modello di collaborazione che - dal 2001 - caratterizza Wikipedia.

## 3.2 Lo sviluppo bottom-up su Wikibase.cloud

Per modellare l'ontologia e gestire i dati, Hyperstage adotta Wikibase.cloud[5], una piattaforma basata su Wikibase, il software di Wikidata. Rispetto ad altri strumenti (es. Protégé),

---

[2] Questa definizione è liberamente tratta da Guarino, Oberle, e Staab 2009.
[3] https://www.wikidata.org/wiki/Wikidata:WikiProject_Performing_arts
[4] https://schema.org/docs/about.html
[5] https://www.wikibase.cloud/

Wikibase.cloud offre semplicità e integrazione con l'ecosistema Wikibase[6], con vantaggi come: facilità d'uso, accessibile anche ai non esperti; trasparenza, con modifiche tracciabili e versioni ripristinabili; gratuità d'uso; efficienza, con costi di sviluppo e manutenzione ridotti per le applicazioni esterne che sfruttano le API di Wikibase. La piattaforma consente di definire l'ontologia mentre si inseriscono dati, caricare metadati tramite API, eseguire query semantiche con SPARQL e creare applicazioni utente (es. portali pubblici) tramite API. Nella pratica, partendo dai metadati forniti dai partner del progetto, le informazioni vengono organizzate e caricate nell'istanza Wikibase.cloud di Hyperstage. Parallelamente si svolge anche un'attività di sviluppo e di manutenzione bottom-up dell'ontologia che integra un mapping semantico a livello di TBox (allineamento dove possibile delle classi e proprietà con Wikidata e Schema.org) e un mapping a livello di ABox (riconciliazione delle istanze tramite identificatori univoci - es. URI Wikidata - per entità che sono già presenti in altre basi di conoscenza).

### 3.3 Le entità principali

Un punto di partenza per il progetto Hyperstage è l'osservazione di una significativa convergenza nella organizzazione dei metadati nelle banche dati che documentano gli spettacoli teatrali:

> The fundamental characteristic that all these databases have in common is that they somehow describe the performing aspect (production, event, performance, listing) and what was performed (play, work, original, original work, script). This can be compared to the way Holden describes the notion that creative activity has two components: the idea and the carrier. In this case: what the performance performs and the performance itself. When the databases represent the performing-aspect, they usually use the production level. (Skaug, e Aalberg 2024, 8)

Si tratta della distinzione tra 'opera' intesa come contenuto intellettuale e le specifiche istanze di 'spettacolo' che mettono in scena quell'opera. Il riferimento a una identità comune migliora l'organizzazione dei dati - in quanto crea connessioni di tipo semantico - e favorisce l'interoperabilità. Un punto di confronto chiave per l'ontologia Hyperstage è l'IFLA Library Reference Model (2017)[7] che definisce l'opera come "the intellectual or artistic content of a distinct creation". Questo modello consolida il lavoro iniziato nel 1998 con FRBR (Functional Requirements for Bibliographic Records). Questo approccio è stato adottato anche dal gruppo *Wikidata:WikiProject Performing Arts*[8].
In Figura 1 si possono trovare le principali classi e proprietà che fondano l'ontologia Hyperstage:

- https://hyperstage.wikibase.cloud/entity/Q1. Spettacolo (theatrical production): "rappresentazione di un evento o serie di eventi quasi identici inclusa la loro produzione complessiva";

---

[6] https://meta.wikimedia.org/wiki/LinkedOpenData/Strategy2021
[7] https://tinyurl.com/IFLALRM2017. Si veda anche LRMoo, versione 1, 2024: https://tinyurl.com/LRMoo2024.
[8] https://www.wikidata.org/wiki/Wikidata:WikiProject_Performing_arts/Data_structure

- https://hyperstage.wikibase.cloud/entity/Q2. Opera creativa teatrale (theatrical creative work): "specifica creazione artistica orientata allo spettacolo teatrale" - il riferimento è all'entità LRM-E2 "the intellectual or artistic content of a distinct creation" ;
- https://hyperstage.wikibase.cloud/entity/Q1445. Agente: superclasse comprendente le sottoclassi Organizzazione (Q5) e Persona (Q6);
- https://hyperstage.wikibase.cloud/entity/Q1355. Risorsa documentale (documentary resource);
- https://hyperstage.wikibase.cloud/entity/Q12. PKB: Classificazione della risorsa secondo la tassonomia PKB;
- https://hyperstage.wikibase.cloud/entity/P6. Messinscena di (staging of): proprietà che collega Q1 a Q2;
- https://hyperstage.wikibase.cloud/entity/P165. Contributi (contribution by): superproprietà che raccoglie le specifiche proprietà indicanti tutti i contributi che un Agente (Q1445) fornisce a Q1 o a Q2;
- https://hyperstage.wikibase.cloud/entity/P110. Documentato da (documented by): proprietà che collega Q1 a Q1355;
- https://hyperstage.wikibase.cloud/entity/P14. Classificazione PKB (PKB classification): proprietà che collega Q1355 a Q12.

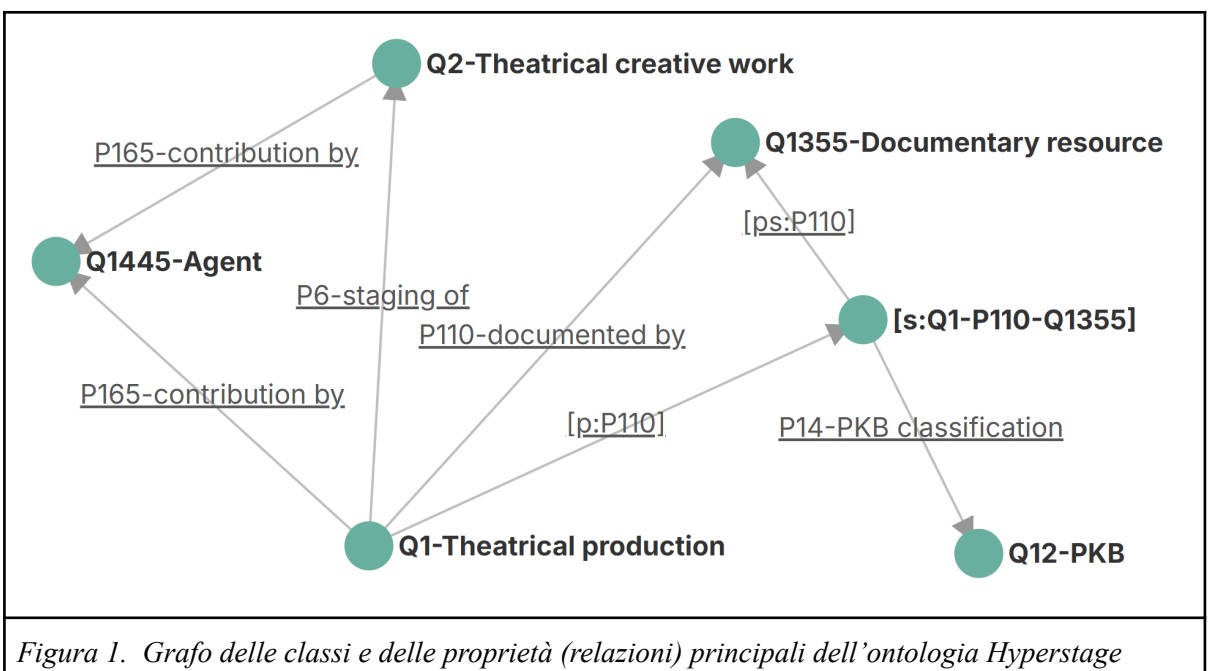

*Figura 1. Grafo delle classi e delle proprietà (relazioni) principali dell'ontologia Hyperstage*

## 3.4 Conflazione e gestione delle ambiguità

Durante il processo di sviluppo bottom-up dell'ontologia Hyperstage, è emersa una problematica significativa concernente la conflazione, ovvero la fusione in un unico item di entità concettualmente distinte. Sebbene tale pratica possa presentare vantaggi in contesti specifici, si rivela controproducente nella costruzione di ontologie complesse come

Hyperstage. La conflazione di istanze, infatti, introduce ambiguità semantiche che compromettono l'usabilità e l'interoperabilità dell'ontologia.

Come argomentato da Guarino, e Welty (2002) nell'ambito della valutazione ontologica con OntoClean, la mancata distinzione tra entità concettualmente separate porta a una perdita di precisione semantica e a potenziali incoerenze all'interno del modello. In particolare, abbiamo riscontrato una potenziale conflazione problematica nella distinzione tra l'opera originale e le sue derivazioni. Queste ultime rappresentano le diverse produzioni che si sono succedute nel tempo.

Per illustrare la complessità della gestione dei metadati relativi alla proprietà 'coreografo (P42)', si consideri il caso del balletto *Giselle*. La coreografia, elemento intrinseco e fondamentale dell'opera coreutica, riveste un ruolo di primaria importanza nella sua definizione.

Nel caso specifico di *Giselle*, capolavoro del periodo romantico debuttato nel 1841, le coreografie originali di Jean Coralli e Jules Perrot hanno esercitato un'influenza significativa sulla storia della danza. Tuttavia, la persistente popolarità dell'opera ha portato, nel corso del tempo, alla creazione di numerose nuove interpretazioni e riallestimenti. Queste produzioni presentano spesso revisioni coreografiche che riflettono l'evoluzione delle tecniche e delle sensibilità artistiche contemporanee.

Tale evoluzione solleva una questione metodologica: come gestire in modo accurato e disambiguo le informazioni relative ai coreografi delle diverse rappresentazioni all'interno di un sistema informativo o di una piattaforma dedicata?

Una semplice attribuzione sistematica di Jean Coralli e Jules Perrot come coreografi di ogni rappresentazione successiva al 1841 condurrebbe a un problema di conflazione. In questo contesto, la conflazione si manifesta come un'attribuzione errata o ambigua di informazioni, in quanto l'associazione esclusiva di Coralli e Perrot oscurerebbe il contributo specifico dei coreografi che hanno apportato naturali rielaborazioni all'opera originale. Tale pratica genererebbe un'informazione distorta e non rappresentativa della reale paternità coreografica delle produzioni successive.

Per garantire una descrizione precisa e accurata della paternità coreografica di *Giselle* nelle sue diverse manifestazioni, è importante fare una chiara distinzione tra la coreografia originale di Coralli e Perrot e i successivi riallestimenti (o adattamenti).

## 3.5 Gerarchizzazione delle produzioni teatrali: un modello per la gestione dei metadati

Al fine di prevenire fenomeni di conflazione e ambiguità, Hyperstage adotta un modello di gerarchizzazione delle opere. Questa decisione si rivela cruciale per mantenere una distinzione netta tra le diverse opere, prevenendo l'erronea fusione tra entità in un unico item. Infatti, attraverso la definizione di livelli gerarchici, è possibile stabilire relazioni chiare e non ambigue, facilitando l'analisi, l'origine e l'evoluzione delle diverse componenti di una produzione teatrale.

Lo schema seguente illustra la struttura gerarchica delle produzioni teatrali, evidenziando le relazioni tra le diverse fasi e versioni delle produzioni teatrali:

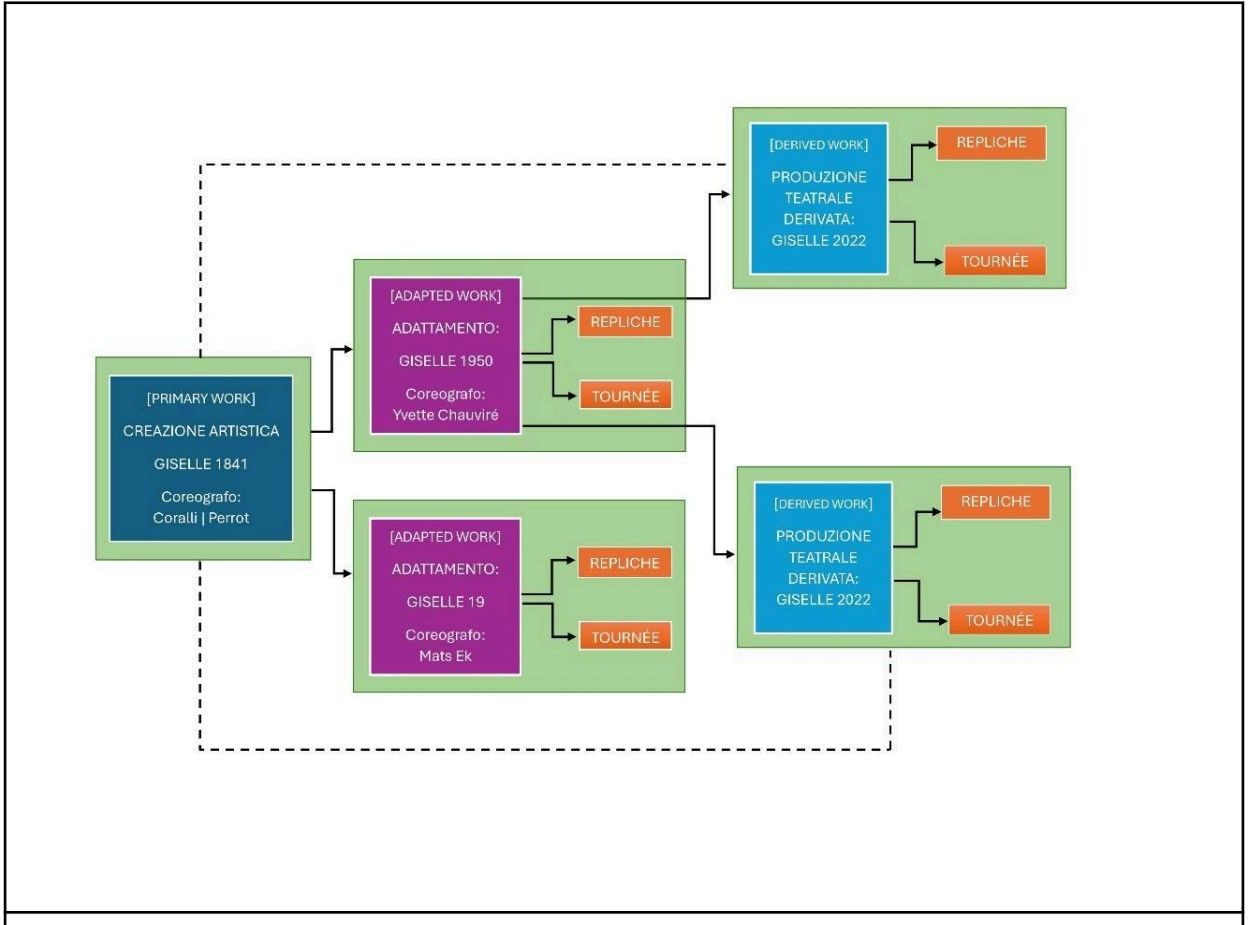

*Figura 2. Schema gerarchico delle produzioni teatrali*

Al fine di stabilire definizioni univoche prima di procedere con la descrizione dello schema gerarchico, è necessario affrontare la complessa nozione di 'opera'. Come altri modelli di catalogazione delle arti performative, Hyperstage si confronta con questa sfida, per la quale il concetto di *Gesamtkunstwerk* (opera d'arte totale) offre un utile parallelismo. Analogamente a Wagner, che teorizzava l'unione di musica, teatro e scenografia in un'unica opera, Hyperstage mira a fornire agli utenti un'esperienza di consultazione unitaria. Tuttavia, la definizione di 'opera' in Hyperstage richiede ulteriori chiarimenti, data la sua intrinseca complessità nelle arti performative, come evidenziato da Skaug, e Aalberg (2024). Un altro interessante punto di riferimento nel panorama scientifico è fornito da Mitsopoulou, Kyprianos e Brattis (2024). In questo lavoro, gli autori sottolineano l'importanza delle strutture gerarchiche per l'organizzazione della conoscenza, un aspetto cruciale anche per Hyperstage. Come evidenziano Mitsopoulou, Kyprianos e Brattis (2024, 6), la tassonomia, uno strumento fondamentale per la classificazione, è intrinsecamente una gerarchia di termini. L'articolo riporta infatti la definizione di tassonomia come "taxonomy as a hierarchy of terms that denote object types", il che implica una struttura teorica di termini organizzata in un grafo, con nodi rappresentati dai termini e archi che ne esprimono le relazioni. Questa struttura si

sviluppa a partire da un singolo termine, la radice della gerarchia, evidenziando come la classificazione gerarchica sia essenziale per ordinare e strutturare le informazioni in modo chiaro e logico, un principio che è stato tenuto presente nello sviluppo di Hyperstage. Per garantire la precisione nella gerarchizzazione di Hyperstage, è necessario chiarire la terminologia fondamentale:

- Opera creativa. Si tratta di un concetto astratto che si riferisce a una specifica opera teatrale 'generatrice', dalla quale si sono sviluppate tutte le altre. Questa 'opera generatrice' rappresenta l'ideazione originale, comprensiva di elementi fondamentali quali il soggetto, il compositore, l'autore, il coreografo. Ad esempio, il balletto *Giselle*, la cui 'opera creativa' è ricondotta al libretto originale di Jules-Henri Vernoy de Saint-Georges, Théophile Gautier e Jean Coralli, con le musiche di Adolphe Adam, coreografia originale di Jean Coralli e Jules Perrot (1841). In Hyperstage, l'opera creativa di *Giselle* funge da punto di origine per tutte le successive messe in scena, produzioni e adattamenti del balletto.
- Produzione. Si intende l'insieme degli spettacoli che condividono un allestimento scenico, costumi, impostazione musicale e impianto coreografico sostanzialmente identici. Questo concetto si riferisce alla collaborazione creativa di un gruppo di artisti e tecnici (regista, scenografo, costumista, musicista, coreografo, interpreti, illuminotecnica ecc.) per realizzare una specifica interpretazione di un'opera.
- Rappresentazione. Con questo termine si intende una specifica messinscena, ovvero la rappresentazione di un'opera teatrale o performativa in un determinato luogo e momento. Ogni spettacolo è un evento unico e irripetibile, anche se fa parte di una stessa produzione.

Alla luce di quanto esposto, è possibile individuare tre livelli di gerarchizzazione:

1) Primo livello. 'Primary work | Creazione Artistica' (in blu scuro): rappresenta la concezione originale (generatrice) della produzione teatrale. È il punto di partenza da cui si diramano tutte le successive elaborazioni e adattamenti.

2) Secondo livello 'Adapted Work | Adattamento' (in viola): si configura come una nuova versione della Creazione Artistica [Primary work], in cui uno o più elementi costitutivi della stessa (come la regia, la scenografia, la coreografia, i costumi) vengono rielaborati da altri artisti o da altri professionisti. L'Adattamento si configura come un'opera autonoma, distinta dalla Creazione Artistica [Primary work] proprio per le modifiche apportate.

3) Terzo livello 'Derived Work | Derivazione' (in celeste): identifica una produzione teatrale che non deriva direttamente dalla Creazione Artistica originale, bensì da un suo Adattamento, pur mantenendo un legame genetico con l'opera originale – nella Fig. 2 si può notare una linea tratteggiata che collega la Primary work con la Derived Work – . La Derivazione presenta ulteriori trasformazioni e rielaborazioni rispetto all'Adattamento da cui trae origine.

Le repliche e le tournée (in arancione) rappresentano le esecuzioni successive alla prima messinscena di una determinata produzione. Le repliche sono le rappresentazioni che si svolgono nella stessa sede, mentre le tournée indicano una serie di rappresentazioni della stessa produzione in diverse località. Sebbene possa sembrare una semplice riproposizione, ogni replica presenta una sua unicità, con variazioni che possono riguardare il cast, gli allestimenti e, soprattutto nel nostro contesto digitale, i materiali ad essa associati (immagini e documenti).

Hyperstage si è confrontato con la necessità di rappresentare digitalmente questa ricchezza di sfumature, portandoci a esplorare due approcci differenti su wikibase.cloud:

1) La creazione di un item specifico per ogni replica: nonostante la chiarezza e la precisione garantite, questo approccio si è dimostrato insostenibile nel lungo periodo, generando una quantità di dati difficilmente gestibile. Unica eccezione è rappresentata dalle produzioni interamente realizzate all'estero, per le quali è stato creato un item specifico per la produzione in tournée.

2) L'inserimento delle proprietà 'repliche (P22)' e 'tournée (P112)' all'interno dell'entità relativa ad una produzione specifica (QXXX): questo secondo approccio si è rivelato più flessibile e sostenibile. Consente di associare le informazioni dettagliate di ogni replica direttamente alla produzione teatrale principale, senza moltiplicare inutilmente gli elementi. Tuttavia, richiede una progettazione accurata delle proprietà e delle relazioni tra i dati, per garantire che tutte le informazioni rilevanti siano catturate e rese accessibili.

Grazie ai qualificatori e ai riferimenti di una specifica dichiarazione[9], è possibile creare una rete di connessioni semantiche che arricchisce significativamente la rappresentazione delle informazioni e facilita la ricerca e l'analisi dei dati.

In sintesi, la gerarchizzazione delle produzioni teatrali, unita a una gestione uniforme di repliche e tournée, si rivela fondamentale per tracciare con precisione l'evoluzione di un'opera. Questo approccio permette di distinguere chiaramente la forma originale, le sue reinterpretazioni, superando le ambiguità nella gestione dei metadati.

## 3.6 Implementazione del modello gerarchico delle produzioni teatrali in Hyperstage

Per i dettagli della gerarchizzazione precedentemente descritta si rinvia alla pagina di hyperstage.wikibase.cloud: https://tinyurl.com/P113-P115-P116. Si rinvia anche alla Fig. 3 dove è rappresentato il grafo che illustra il modello implementato in Hyperstage per strutturare la gerarchia delle produzioni teatrali di Giselle.

---

[9] Per il Wikibase data model: https://www.mediawiki.org/wiki/Wikibase/DataModel

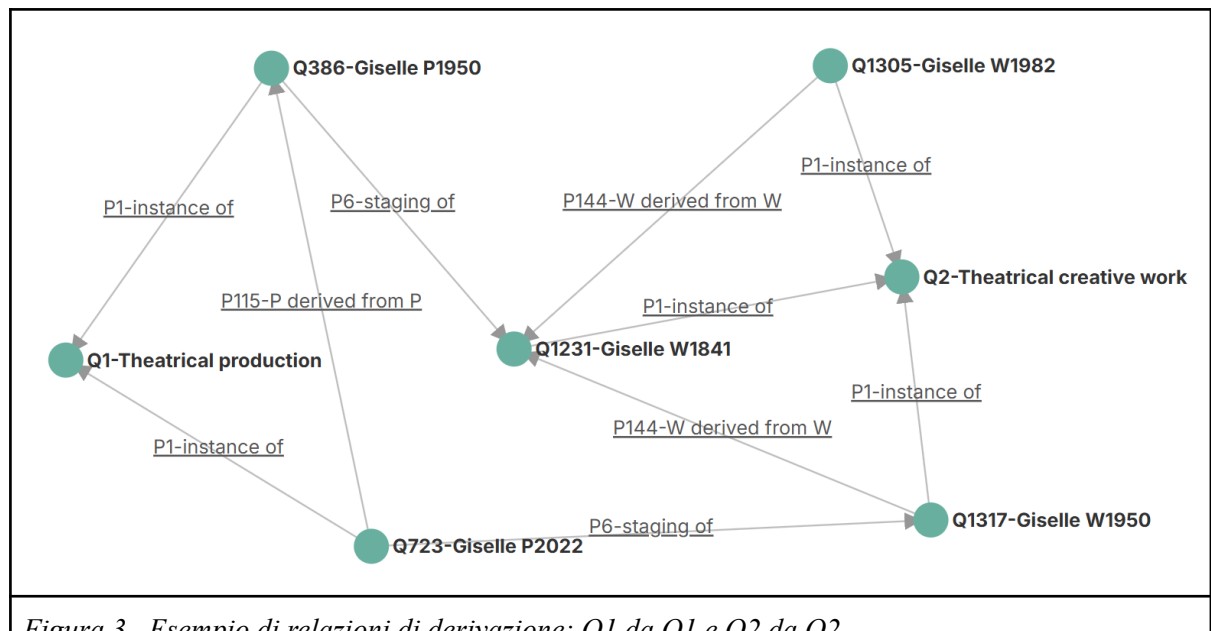

*Figura 3. Esempio di relazioni di derivazione: Q1 da Q1 e Q2 da Q2*

## 4. Accesso alle risorse digitali: il sistema di classificazione PKB

### 4.1 Gestione delle risorse digitali in Hyperstage (wikibase.cloud)

Una singola produzione teatrale può generare e aggregare una quantità considerevole di risorse digitali. In wikibase.cloud abbiamo sperimentato diverse metodologie per la gestione delle risorse digitali, optando infine per un approccio basato sulla categorizzazione delle risorse in base alle loro caratteristiche intrinseche. Questo approccio prevede la suddivisione in due grandi categorie:

1) Foto di scena, foto delle prove, foto del backstage (con interpreti) e documenti dello spettacolo (locandine, informative, brochure, programma di sala ecc.).
2) Bozzetti di costume, bozzetti di scena, foto del costume realizzato, foto della scena realizzata.

Tale distinzione è motivata dalla diversa natura delle risorse documentali. La prima categoria, infatti, raccoglie elementi non corredati da dati tecnici dettagliati, mentre la seconda, comprende risorse con informazioni tecniche associate.

Hyperstage ha l'obiettivo di massimizzare il valore delle risorse digitali, creando connessioni significative tra di esse. Per questo obiettivo è stata sfruttata la capacità di Wikibase di precisare le dichiarazioni con qualificatori e riferimenti, consentendo a Hyperstage di arricchire la comprensione e la reperibilità delle sue risorse digitali. Un esempio chiave di questa capacità si trova nell'utilizzo della proprietà P31 'documentato da ( con codominio[10] stringa)'. Anche se superficialmente potrebbe sembrare una semplice stringa, Hyperstage sfrutta appieno il suo potenziale attraverso l'impiego di numerosi qualificatori. Questi qualificatori trasformano una semplice indicazione in un ricco insieme di metadati contestuali. Essi permettono di collocare la risorsa digitale in modo preciso nel tempo e nello spazio, fornendo dettagli fondamentali come: informazioni sul fotografo (identificando

---

[10] Codominio o rdfs:range  è l'insieme di valori che una proprietà può assumere:
https://www.w3.org/TR/rdf-schema/#ch_range

l'autore dell'immagine); dettagli sul materiale utilizzato (specificando la tecnica o il supporto impiegato es. pellicola, digitale, tipo di carta); contesto temporale (EDTF); localizzazione precisa; persone ritratte (specificando e collegando il nome dell'interprete alla risorsa digitale); informazioni sullo spettacolo (come l'atto, la scena o il momento specifico dello spettacolo documentato) ecc.

Per gestire in modo coerente le informazioni associate a risorse digitali particolarmente complesse, abbiamo creato la proprietà P110 'documentato da (con codominio elemento)'. Queste risorse includono: bozzetti di costume, bozzetti di scena, fotografie del costume realizzato e fotografie della scena realizzata. Per semplificare la struttura dei dati, le abbiamo raggruppate nella proprietà 'sottoclasse di (P2)': 'risorsa documentale (Q1355)'.

Per ognuna di queste risorse è stato creato un elemento specifico. Ad esempio, per il bozzetto di scena intitolato *La clinica*, realizzato per lo spettacolo di Dino Buzzati al Piccolo di Milano (1953), abbiamo creato l'elemento (Q1333)[11]. A questo elemento possiamo associare tutte le dichiarazioni e i relativi qualificatori, senza rischio di ambiguità. Questo sistema ci permette di raccogliere tutte le informazioni necessarie, incluse quelle utili per confrontare e collegare le diverse risorse tra loro, generando così una sorta di scheda catalografica.

Le proprietà utilizzate come qualificatori in 'documentato da (P31)' e come dichiarazioni in 'documentato da (risorsa) (P110)' sono elencate in questa pagina su hyperstage.wikibase.cloud: https://tinyurl.com/hyperstageprd; in questa altra pagina il dettaglio delle differenze: https://tinyurl.com/DRP31-P110.

## 4.2 Un sistema interconnesso

Le proprietà mostrate nella tabella al link precedente sono fondamentali per trasformare le risorse documentali in un sistema realmente interconnesso. Queste proprietà permettono di creare legami significativi tra i diversi elementi, andando oltre la semplice archiviazione. Ad esempio, è possibile collegare una foto di un bozzetto di costume alla fotografia del costume realizzato e, successivamente, all'immagine dell'attore che indossa quel costume in scena. Questa connessione non solo documenta le singole risorse, ma ricostruisce l'intero processo creativo legato a uno specifico elemento dello spettacolo.

Le potenzialità di questa interconnessione sono molteplici. Un utente può facilmente risalire a tutti gli interpreti che hanno indossato uno specifico costume nel corso delle rappresentazioni, oppure confrontare un modellino di scena con le fotografie della scena effettivamente realizzata.

Il valore aggiunto del nostro sistema risiede proprio in questa capacità di mettere in relazione le risorse digitali. Un ricercatore o un appassionato può esplorare una specifica produzione teatrale e osservare l'evoluzione delle risorse ad essa associate nel tempo, tracciando per esempio le modifiche apportate ai costumi tra le diverse repliche o le variazioni scenografiche.

Le proprietà P31 'documentato da ( con codominio stringa)' e P110 'documentato da (con codominio risorsa) ' arricchite rispettivamente dai qualificatori e dalle dichiarazioni, sono un

---

[11] https://hyperstage.wikibase.cloud/wiki/Item:Q1333

potente strumento in questo processo, permettendo di contestualizzare profondamente ogni elemento e di svelare relazioni altrimenti nascoste tra le diverse risorse.

**4.3 Ordinare la 'scena digitale': l'efficacia della classificazione PKB**

Nell'ambito della documentazione delle produzioni teatrali, un elemento cruciale per garantire un'organizzazione ottimale e una facile reperibilità delle risorse documentali è rappresentato dal sistema di classificazione PKB. La classificazione PKB è stata mappata in un modello ontologico espresso in formato RDF e reso interrogabile tramite query SPARQL. Ciò significa che a ogni tipo specifico di risorsa documentale è attribuito un codice alfanumerico grazie al quale è possibile identificare in maniera univoca la tipologia della risorsa. La sua architettura si fonda su due macro-categorie principali: 'pictures' (immagini) e 'documents' (documenti). All'interno di queste, come descritto da Gavrilovich (2020), la classificazione PKB struttura le risorse in base a tre fasi fondamentali che ne determinano il processo creativo e i riscontri, permettendo di collegare materiali eterogenei all'interno di un flusso produttivo:

1. Concezione
2. Realizzazione
3. Post-produzione

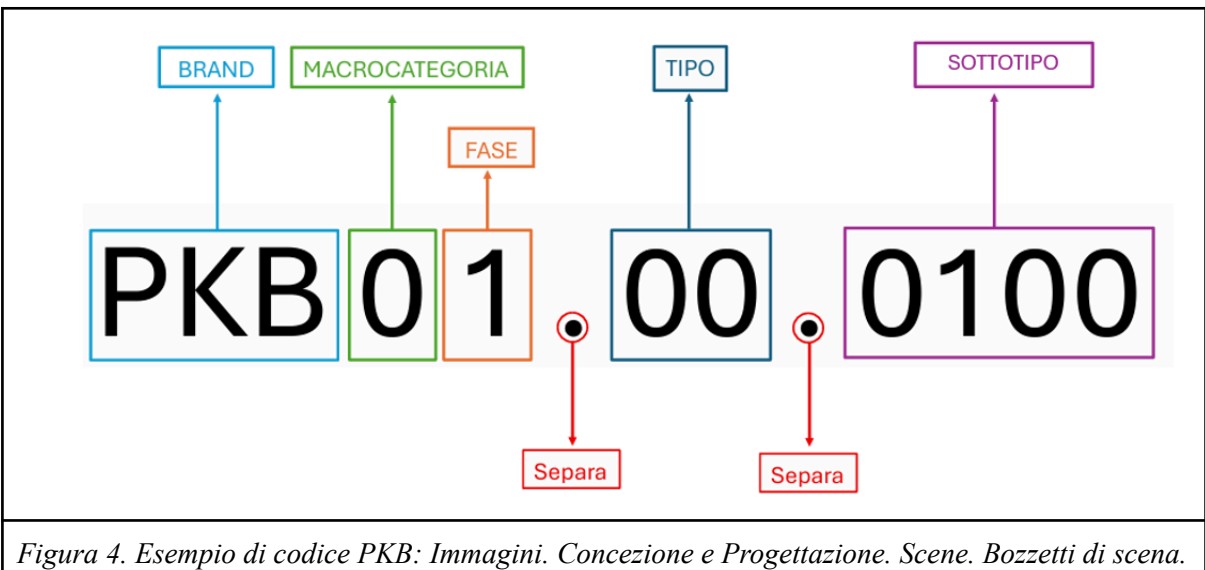

*Figura 4. Esempio di codice PKB: Immagini. Concezione e Progettazione. Scene. Bozzetti di scena.*

Questa tripartizione temporale permette di contestualizzare accuratamente ogni risorsa. Un codice PKB (Fig. 4) è costituito da:

- 'PKB' è un acronimo che serve a facilitare l'identificazione del brand (in questo caso, il sistema di classificazione stesso) e a ridurre l'ambiguità.
- Le due cifre '01' successive identificano - con la prima cifra, che può assumere valore 0 oppure 1  le due macro–categorie e - con la seconda cifra, che può assumere i valori 1,2,3 - le rispettive 3 fasi del processo creativo:
  - ➢ 01 (IMMAGINI, concezione)

> ➢ 02 (IMMAGINI, realizzazione)
> ➢ 03 (IMMAGINI, post-produzione)
> ➢ 11 (DOCUMENTAZIONE, concezione)
> ➢ 12 (DOCUMENTAZIONE, realizzazione)
> ➢ 13 (DOCUMENTAZIONE, post-produzione)

- Il punto separa la macrocategoria/fase dal tipo.
- Le due cifre a seguire '00', delimitate anch'esse dal punto, identificano il tipo di risorsa ossia: scene, personaggi attrezzi di scena, registrazioni video, foto, immagini, manifesti ecc..
- Il secondo punto separa il tipo dal sottotipo.
- Le quattro cifre successive indicano:
  > ➢ la prima e la seconda, il sottotipo 1 (le due cifre permettono di ospitare valori che vanno da 00 a 99). Questo primo livello definisce in modo più preciso la risorsa rispetto alla categoria superiore (tipo).
  > ➢ la terza, il sottotipo 2, aggiunge un ulteriore livello di dettaglio alla specificazione fornita dal sottotipo 2.
  > ➢ la quarta, il sottotipo 3, definisce in modo ancora più granulare e dettagliato la risorsa.

Queste quattro cifre lavorano in sequenza per restringere progressivamente il campo di definizione della risorsa, passando da una classificazione più ampia a una sempre più dettagliata.

La classificazione completa si trova in questa pagina di hyperstage.wikibase.cloud: https://tinyurl.com/classpkb.

## 5. Conclusioni

I risultati intermedi del progetto Hyperstage confermano l'adeguatezza della scelta tecnologica di Wikibase.cloud  per la gestione della ricostruzione semantica delle produzioni teatrali attraverso la raccolta e l'elaborazione di risorse (metadati e risorse digitali documentali)  provenienti dagli archivi dei partner del progetto. In particolare il fatto che Wikibase consenta uno sviluppo bottom-up dell'ontologia ha permesso di sperimentare rapidamente - con dati reali - la sostenibilità di soluzioni che provengono sia da database che documentano spettacoli teatrali (come ad esempio IbsenStage) sia da proposte che provengono anche dal mondo dei dati bibliografici (IFLA LRM).
Il modello si struttura a partire da un nucleo essenziale di entità e dalle loro relazioni. Le entità principali: Q1445 'Agente', Q1 'Spettacolo (Theatrical production)', Q2 'Opera creativa teatrale (Theatrical creative work)', Q1355 'Risorsa documentale (Documentary resource)', Q12 'Classificazione PKB'.  Oltre alla relazione P6 'messinscena di (staging of)' tra Q1 'Spettacolo' e  Q2 'Opera creativa teatrale' che consente il riferimento ad una identità comune al di là delle specifiche istanze di Spettacolo, sono state prese in considerazione le relazioni di tipo gerarchico:  P115 'spettacolo derivato da spettacolo' , una relazione tra entità Q1 'Spettacolo';   P113 'opera teatrale derivata da opera teatrale', una relazione tra entità Q2 'Opera creativa teatrale'.

Le funzionalità offerte dai qualificatori (qualifiers)  e dai riferimenti (references) previsti nel Wikibase data model sono state sfruttate da Hyperstage in molti contesti, tra questi:  la gestione delle repliche e la relazione tra Spettacolo e Risorsa documentale.  Senza i qualificatori anche esprimere correttamente con la sintassi RDF[12]  che ad esempio «Carla Fracci ha interpretato Giselle nello Spettacolo Giselle al Teatro alla Scala nel 1971» risulterebbe estremamente complesso[13].

Infine la classificazione PKB che organizza la documentazione in formato digitale dello Spettacolo ha trovato il suo posto all'interno dell'ontologia: ogni simbolo di classificazione è definito come istanza della classe 'classificazione PKB'.

Le prospettive di lavoro future comprendono  l'introduzione di meccanismi di controllo dell'inserimento dei dati basati su ShEx[14] e  il potenziamento del mapping sia a livello di ABox che di TBox.  Un obiettivo critico è definire un framework per la manutenzione collaborativa dell'ontologia anche dopo il termine del progetto.

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
