# OpenReview forum: "An ontology for Italian theatrical cultural heritage on wikibase.cloud"
_wikimedia.it/Wikidata_and_Research/2025/Conference — WD&R Paper_

### Official Review · ~Silvia_Bruni1 · 2025-01-10
**An ontology for theatrical productions in Wikidata: enhancig the organization and accessibility of data related to the performing arts**

**Originality:** 5
**Impact:** 5
**Confidence:** 4

**Review:**

The project, working on an ontology, is undoubtedly very innovative within the context of Wikidata. It addresses, moreover, a set of highly complex data concerning theatrical performances. These digital objects, as emphasized in the abstract, refer to different phases of production: conception, staging, and documentation. The proposed and tested solutions certainly deserve to be illustrated and showcased. It is noteworthy that an ontology concerning works that are inherently ephemeral holds added value in terms of data preservation. The project aims to transcend the traditional limits of theatrical documentation, which are not entirely resolved by bibliographic models such as IFLA LRM. The project aims to be a solution for the enhancement of digital theatrical archives through various innovative strategies and advanced technologies.

**Compliance:**

5

**Scientific Quality:**

5

---

### Official Review · ~Carlo_Bianchini1 · 2025-01-13
**A Wikibase instance for the collection, organization, and valorization of metadata related to theatrical productions**

**Originality:** 5
**Impact:** 5
**Confidence:** 4

**Review:**

The proposal investigates the possibility to create a dataset that leverage on the distinction between theatrical creative work and theatrical production, and offers a methodologicallty innovative and relevant solution for the collection, organization, and valorization of metadata related to theatrical productions. Of great interest the teonomy based on the three different categories of a) conception, b) staging, and c) post-production documentation.

**Compliance:**

5

**Scientific Quality:**

5

---

### Decision · Program_Chairs · 2025-02-05

Accept (Paper)